# The impact of chronic comorbidities at the time of breast cancer diagnosis on quality of life, and emotional health following treatment in Canada

**Jasleen Arneja****, Jennifer D. Brooks***

Dalla Lana School of Public Health, University of Toronto, Toronto, Ontario, Canada

* jennifer.brooks@utoronto.ca

## Abstract

### Introduction

Advances in breast cancer screening and treatment have led to an increasing number of breast cancer survivors. The objective of this study was to determine the impact of comorbidities on self-reported quality of life (QOL) and emotional health following a breast cancer diagnosis and treatment.

### Methods

Women with a personal history of breast cancer (N = 3,372) were identified from the cross-sectional Canadian Partnership Against Cancer (CPAC) Experiences of Cancer Patients in Transitions Survey. Multinomial (nominal) logistic regression was used to estimate odds ratios (OR) and 95% confidence intervals (CI) for the relationship between burden of comorbidities and overall QOL and emotional health (very poor/poor, fair, good, very good).

### Results

Of the 3,372 participants, 57% reported at least one chronic condition at the time of breast cancer diagnosis. As the number of chronic conditions at diagnosis increased, the odds of reporting worse quality of life and emotional health following treatment also increased. Specifically, compared to women reporting very good QOL, for each additional chronic condition, women reported significantly higher odds of reporting good (OR = 1.22, 95% CI: 1.12, 1.32), fair (OR = 1.76, 95% CI: 1.58, 1.96), or poor/very poor (OR = 2.31, 95% CI: 1.86, 2.88) QOL. Similarly, for each additional comorbidity, women reported significantly higher odds of reporting good (OR = 1.17, 95% CI: 1.07, 1.28), fair (OR = 1.63, 95% CI: 1.46, 1.82), or poor/very poor (OR = 2.17, 95% CI: 1.81, 2.60) emotional health, relative to very good emotional health.

### Conclusion

Breast cancer survivors coping with a high comorbidity burden experience worse overall QOL and emotional health following treatment. This highlights the importance of integrating

**Data Availability Statement:** Data are publicly available via the Canadian Partnership Against Cancer, at the following URL: https://www.

systemperformance.ca/transitions-study/
transition-study-questions/.

**Funding:** This study was funded in part by a CIHR Project grant (RN312225-376411) awarded to J.B. No additional external funding was received for this study. The funders had no role in study design, data collection, and analysis, decision to publish, or preparation of the manuscript.

**Competing interests:** The authors have declared that no competing interests exist.

information on comorbidities into survivorship care to improve the experience and overall outcomes of patients with complex needs.

## Introduction

Breast cancer is the most common cancer diagnosed among North American women, with 1 in 8 expected to develop the disease in their lifetime [1]. Advances in breast cancer screening and treatment have led to an increasing number of breast cancer survivors. As 5-year net survival for breast cancer in Canada approaches 90% [1], addressing the needs of these women and emphasizing the improvement of overall quality of life (QOL) is increasingly important.

In Canada, 83% of breast cancer cases occur in women over 50 years of age [2, 3], therefore the majority of survivors are older women. Chronic comorbidities (e.g., pulmonary disease, and dementia) are common in 20–35% of breast cancer patients, and their prevalence can be as high as 86% in patients aged 65 and above [4]. Presence of comorbidities at the time of diagnosis can negatively impact survival after breast cancer diagnosis, especially in these women [5, 6]. Prior work has shown that breast cancer survivors with any comorbidity have a significantly higher all-cause and breast cancer-specific mortality [7]. Furthermore, improvements in breast cancer survival witnessed over the past three decades have not been observed in breast cancer patients with severe comorbidities [7, 8].

QOL is a multidimensional measure encompassing physical, mental, social, economic, and spiritual aspects. As such, QOL is impacted by a multitude of factors. Socioeconomic factors, including social isolation [9], low socioeconomic status [9, 10], and lower neighborhood or household level income [11, 12] have been found to be associated with poorer QOL among breast cancer survivors. Conversely, greater social support [13], and support satisfaction [14] have a positive impact on QOL.

In breast cancer survivors, QOL is also inextricably linked to survival. In a cohort of women over 65 years of age with early-stage breast cancer, health related QOL measures predicted 10-year mortality independently of traditional breast cancer prognostic variables [4], suggesting that interventions aimed at improving physical function, mental health, and social support might improve both health related QOL and survival. Further, young breast cancer survivors (i.e., age ≤45 years) frequently report worse QOL [15], often stemming from menopausal symptoms, problems with relationships, and sexual functioning [16]. Premenopausal breast cancer survivors have been known to experience decreased emotional wellbeing, increased anxiety and body image issues [17].

In order to improve quality of life and overall survival among breast cancer survivors in Canada, the impact of comorbidity burden on both QOL and emotional well-being warrants examination [18]. The objective of this study was to leverage a large pan-Canadian survey of cancer survivors to determine the association between the comorbidity burden—as captured by the number of chronic conditions at the time of diagnosis, and both self-reported QOL and emotional health following treatment for breast cancer.

## Materials and methods

### Data sources

The data for this study were obtained from the Experiences of Cancer Patients in Transitions Study (Transitions Study), a cross-sectional study of adult cancer survivors over the age of 18 years, conducted by the Canadian Partnership Against Cancer (CPAC). The goal of this study

was to better understand challenges related to cancer survivorship. A comprehensive description of survey methods, development, validation, and dissemination have been published elsewhere [19, 20]. Briefly, in 2016, 40,790 survey packages were mailed to adolescent and adult cancer patients identified through the provincial cancer agencies/registries of 10 Canadian provinces. This included survivors of breast, colorectal, prostate, melanoma, and hematological cancers who completed treatment (chemotherapy, radiation therapy, surgical treatment, or a combination of these therapies) in the previous 1–3 years. Data were collected in parallel across the 10 provinces, with the recruitment period ranging from 8 to 19 weeks for different provinces. The study population was selected through probability sampling. A sampling error margin of ±5% for the 95% CI was used to calculate the sample size for each disease site and province, assuming a response rate of 30% [19]. In smaller provinces, where the desired precision was not achieved, all eligible survivors were interviewed [19], which—combined with recruitment through provincial cancer agencies—minimized selection bias. In total, 13,319 responses were received, corresponding to a response rate of 33%.

The data used in this study are in the public domain and available through the CPAC website [21]. The authors were not involved in any aspect of the original study and did not have access to any identifying information associated with the data. This study was from ethics review. Results are reported in accordance with the STROBE statement [22], (see STROBE checklist, S1 File).

**Exposure and outcomes.** The Transitions Study survey captured personal and demographic characteristics; needs as a cancer survivor (physical, emotional, informational, and practical); and enablers and barriers of these needs being met. It also collected data on the prevalence of chronic conditions at the time of breast cancer diagnosis. This included: 1) arthritis, osteoarthritis, or other rheumatic disease, 2) cardiovascular or heart condition, hypertension or high blood pressure, 3) chronic kidney disease, 4) diabetes, 5) osteoporosis, 6) respiratory diseases (e.g., asthma or COPD—chronic obstructive pulmonary disease) and 7) mental health issues (e.g., depression or anxiety), with a free text field for other chronic conditions. The number of chronic conditions a participant reported was defined as the sum of the 6 most prevalent chronic conditions (>8% prevalence in breast cancer survivors). Specifically, these included: arthritis, cardiovascular disease, diabetes, osteoporosis, respiratory illness, and mental health issues.

Overall QOL was assessed at the time of the survey i.e. after treatment completion, using the following question: "How would you describe your overall quality of life today?" Emotional health was assessed similarly using the question, "In general, would you say your emotional health is. . .". Both variables were assessed using a Likert scale of very poor, poor, fair, good, very good. The Transitions Study collected data on overall QOL, a broad and multidimensional concept. Prior literature [23, 24] has focused specifically on health related quality of life, which is a patient reported outcome, usually measured using multi-item surveys such as the European Organization for Research and Treatment of Cancer (EORTC) Quality of Life Questionnaire Core 30 (QLQ-C30), or the Short Form Health Survey (SF-36 for the 36 item version of the survey). The QOL measure included in the CPAC Transitions Study questionnaire is comparable the global QOL measure from the EORTC QLQ-C30, which has been validated for use in breast cancer patients [25–27].

## Study population

Of the 13,319 Transitions Study survey respondents, N = 3,729 women over the age of 30 years reported having a personal history of breast cancer. This was assessed by asking women about their most recent cancer diagnosis. For 330 women, (9.8%) the index cancer was not their first

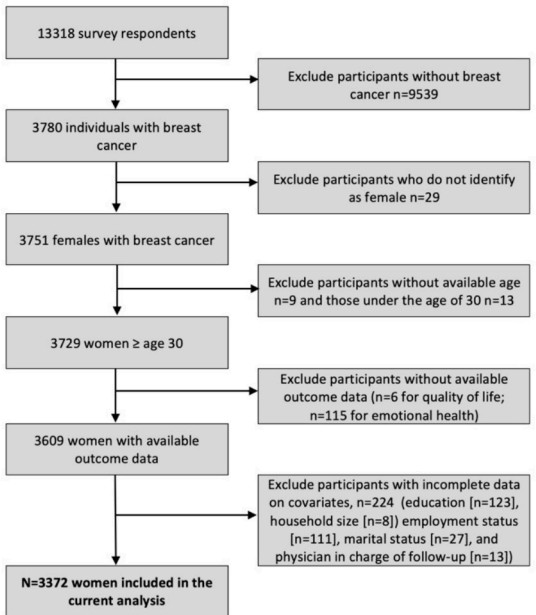

**Fig 1. Sample size: Flow diagram of Transitions Study survey respondents included in the analysis.**

cancer diagnosis. Individuals with missing data on age at data collection (N = 9), QOL (N = 6), and emotional health (N = 115) were excluded. Women with missing data on covariates (education [n = 123, 3.4%], household size [n = 8, 0.2%], employment status [n = 111, 3.1%], marital status [n = 27, 0.8%], and whether a physician was in charge of their follow-up care [n = 13, 0.4%]), were also excluded leaving a final study population of 3,372 women for the current analysis (Fig 1).

## Statistical analysis

Generalized logit models were used to estimate odds ratios (OR) and 95% confidence intervals (CI) for the relationship between number of chronic conditions (continuous) and self-reported QOL and emotional health (very poor/poor, fair, good, very good). Models were adjusted for *a priori* selected variables, including, age, education, employment status, marital status, household size, and whether participants had a healthcare provider such as a nurse or physician in charge of overseeing their follow up care. Variable coding details are presented in S1 Table. Statistical significance was determined at an alpha level of 0.05. All analyses were conducted using SAS version 9.4 (Cary, NC).

## Results

Demographic and physical health characteristics of the study population are reported in Table 1. Most respondents were between 65–74 years of age (32%), had a post-secondary degree (57%) and were married or partnered (71%). More than half of the participants were retired (55%) and lived in a two-person household (56%). Almost all participants had a physician in charge of overseeing their follow-up care (97%). On average, participants had one chronic condition (interquartile range 0–2), and most participants reported one or more

**Table 1. Demographic characteristics of the breast cancer survivors in the Transitions Study, N = 3,372.**

| | N (%)[a] |
|---|---|
| **DEMOGRAPHIC CHARACTERISTICS** | |
| **Age (years)** | |
| 30–34 | 34 (1.0) |
| 35–44 | 142 (4.2) |
| 45–54 | 555 (16.5) |
| 55–64 | 984 (29.2) |
| 65–74 | 1,078 (32.0) |
| 75–84 | 473 (14.0) |
| ≥ 85 | 106 (3.1) |
| **Education** | |
| High School or Less | 1,206 (35.8) |
| Post-Secondary Degree | 1,931 (57.3) |
| Graduate School | 235 (7.0) |
| **Income** | |
| < $25,000 | 439 (13.0) |
| $25,000 to < $50,000 | 729 (21.6) |
| $50,000 to < $75,000 | 549 (16.3) |
| $75,000 or more | 847 (25.1) |
| Missing | 808 (24.0) |
| **Employment Status** | |
| Employed (full time/part time) | 1,133 (33.6) |
| Unemployed, homemaker, student, or on paid sick leave | 371 (11.0) |
| Retired | 1,868 (55.4) |
| **Marital Status** | |
| Single | 198 (5.9) |
| Married, or partnered | 2,390 (70.9) |
| Divorced, separated, or widowed | 784 (23.3) |
| **Household Size** | |
| 1 (Live Alone) | 735 (21.8) |
| 2 | 1,898 (56.3) |
| 3 | 374 (11.1) |
| 4 | 242 (7.2) |
| 5 or More | 123 (3.7) |
| **Physician in Charge of Follow-up** | |
| Yes | 3,263 (96.8) |
| No or Unsure | 109 (3.2) |
| **CHRONIC CONDITIONS** | |
| **Arthritis** | |
| No | 2,297 (68.1) |
| Yes | 1,075 (31.9) |
| **Cardiovascular Disease** | |
| No | 2,522 (74.8) |
| Yes | 850 (25.2) |
| **Diabetes** | |
| No | 3,092 (91.7) |
| Yes | 280 (8.3) |
| **Osteoporosis** | |

(*Continued*)

**Table 1.** (Continued)

| | |
|---|---|
| No | 3,086 (91.5) |
| Yes | 286 (8.5) |
| **Respiratory Diseases** | |
| No | 3,086 (91.5) |
| Yes | 286 (8.5) |
| **Mental Health Issues** | |
| No | 2,954 (87.6) |
| Yes | 418 (12.4) |
| **Number of Chronic Conditions** | |
| *Median (Interquartile Range)* | 1 (0, 2) |
| 0 | 1,455 (43.1) |
| 1 | 1,015 (30.1) |
| 2 | 604 (17.9) |
| 3 | 229 (6.8) |
| 4 | 61 (1.8) |
| 5 | 7 (0.2) |
| 6 | 1 (0.0) |

Abbreviations: N = number.

[a]Percentages may add up to 100% in some cases due to rounding.

chronic condition at the time of breast cancer diagnosis (57%). The most prevalent chronic condition was arthritis (32%), followed by cardiovascular disease (25%). Notably, women with fewer comorbidities (<4) tended to receive more treatments than those with a higher burden of comorbidities ($\geq$ 4). This was particularly true for chemotherapy (45% vs 35% for women with <4 vs. $\geq$ 4 comorbidities.

Most participants rated their QOL as very good (40%) or good (44%); with only 14% of participants reporting a fair QOL, and 2% reporting a poor or very poor QOL (Table 2). As women's reported number of chronic conditions increased, their odds of reporting worse QOL also increased. Specifically, with each additional reported condition, women had significantly higher odds of reporting good (OR = 1.22, 95% CI: 1.12, 1.32), fair (OR = 1.76, 95% CI: 1.58, 1.96), and poor/very poor QOL (OR = 2.31, 95% CI: 1.86, 2.88), as compared to very good QOL (Table 2). Unadjusted estimates of the relationship between the number of comorbid conditions and QOL are provided in S2 Table.

Higher levels of education were consistently associated with a significantly lower odds of reporting worse QOL (OR = 0.79, 95% CI: 0.70, 0.91 for good, OR = 0.62, 95% CI: 0.51, 0.76 for fair, and OR = 0.60, 95% CI: 0.38, 0.95 for poor/very poor QOL, relative to very good QOL). Unemployed women, those on paid sick leave, and homemakers report worse QOL (OR = 1.62, 95% CI: 1.22, 2.17 for good, OR = 4.33, 95% CI: 3.01, 6.22 for fair, and OR = 8.83, 95% CI: 3.80, 20.49 for poor/very poor QOL, relative to very good QOL), relative to employed women. Being married or partnered was also associated with a lower odds of reporting worse QOL, although this association was only significant for reporting fair versus very good QOL (OR = 0.38, 95% CI: 0.24, 0.59). Lastly not having a physician in charge of overseeing follow-up care was associated with an increased odds of reporting worse QOL (OR = 1.93, 95% CI:1.19, 3.13 for good, OR = 2.09, 95% CI: 1.13, 3.88 for fair, and OR = 5.43, 95% CI: 2.15, 13.73 for poor/very poor QOL, relative to very good QOL) (Table 3). Unadjusted estimates of the relationship between the number of comorbid conditions and emotional health are provided in S2 Table.

**Table 2. Association between the number of comorbid conditions and quality of life among breast cancer survivors in the Transitions Study using multinomial logistic regression.**

| | OR (95% CI)[a] | | |
| --- | --- | --- | --- |
| | **Good** | **Fair** | **Poor/Very Poor** |
| | **N = 1486 (44.1%)** | **N = 472 (14.0%)** | **N = 65 (1.9%)** |
| **Comorbid Conditions[b]** | 1.22 (1.12, 1.32) | 1.76 (1.58, 1.96) | 2.31 (1.86, 2.88) |
| **Age** | 1.13 (1.03, 1.24) | 1.04 (0.90, 1.19) | 0.86 (0.63, 1.16) |
| **Education** | 0.79 (0.70, 0.91) | 0.62 (0.51, 0.76) | 0.60 (0.38, 0.95) |
| **Household Size** | 1.06 (0.96, 1.17) | 1.10 (0.96, 1.26) | 0.66 (0.43, 1.02) |
| **Employment** | | | |
| Employed | REF | REF | REF |
| Unemployed/Paid Sick Leave | 1.62 (1.22, 2.17) | 4.33 (3.01, 6.22) | 8.83 (3.80, 20.49) |
| Retired | 0.87 (0.70, 1.08) | 0.96 (0.69, 1.34) | 1.49 (0.63, 3.57) |
| **Marital Status** | | | |
| Single | REF | REF | REF |
| Married/Partnered | 0.79 (0.56, 1.12) | 0.38 (0.24, 0.59) | 0.55 (0.21, 1.46) |
| Separated/Divorced/Widowed | 0.84 (0.58, 1.22) | 0.82 (0.51, 1.31) | 0.54 (0.20, 1.47) |
| **Physician in Charge of Follow-up** | 1.93 (1.19, 3.13) | 2.09 (1.13, 3.88) | 5.43 (2.15, 13.73) |

Abbreviations: N = number, OR = odds ratio, REF = reference category.

[a]Models were adjusted for age, education, household size, employment status, marital status, and whether a physician was in charge of patient's follow-up (coding details provided in S1 Table). The reference category was women reporting very good QOL (N = 1349, 40.0%).

The majority of women reported either very good (29%) or good (50%) emotional health; with only 18% and 4% reporting fair and poor/very poor emotional health, respectively (Table 3). As the number of chronic conditions increased, the odds of reporting worse

**Table 3. Association between the number of comorbid conditions and emotional health among breast cancer survivors in the Transitions Study using multinomial logistic regression.**

| | OR (95% CI)[a] | | |
| --- | --- | --- | --- |
| | **Good** | **Fair** | **Poor/Very Poor** |
| | **N = 1668 (49.5%)** | **N = 615 (18.3%)** | **N = 124 (3.7%)** |
| **N Comorbid Conditions** | 1.17 (1.07, 1.28) | 1.63 (1.46, 1.82) | 2.17 (1.81, 2.60) |
| **Age** | 0.97 (0.88, 1.08) | 0.83 (0.72, 0.94) | 0.72 (0.57, 0.91) |
| **Education** | 0.79 (0.69, 0.91) | 0.70 (0.58, 0.84) | 0.62 (0.43, 0.88) |
| **Household Size** | 0.95 (0.86, 1.06) | 1.10 (0.97, 1.25) | 0.80 (0.62, 1.03) |
| **Employment** | | | |
| Employed | REF | REF | REF |
| Unemployed/Paid Sick Leave | 1.54 (1.11, 2.15) | 2.75 (1.89, 3.99) | 8.01 (4.53, 14.17) |
| Retired | 0.87 (0.69, 1.09) | 0.91 (0.67, 1.24) | 0.73 (0.39, 1.35) |
| **Marital Status** | | | |
| Single | REF | REF | REF |
| Married/Partnered | 1.03 (0.71, 1.50) | 0.70 (0.45, 1.10) | 0.55 (0.26, 1.17) |
| Separated/Divorced/Widowed | 1.11 (0.75, 1.64) | 0.99 (0.62, 1.61) | 1.04 (0.48, 2.27) |
| **Physician in Charge of Follow-up** | 2.29 (1.29, 4.07) | 2.38 (1.22, 4.67) | 6.66 (2.89, 15.35) |

Abbreviations: N = number, OR = odds ratio, REF = reference category.

[a]Models were adjusted for age, education, household size, employment status, marital status, and whether a physician was in charge of patient's follow-up (coding details provided in S1 Table). The reference category was women reporting very good emotional health (N = 965, 28.6%).

emotional health increased. Specifically, compared to women reporting very good emotional health, for each additional chronic condition, women had significantly higher odds of reporting good (OR = 1.17, 95% CI: 1.07, 1.28), fair (OR = 1.63, 95% CI: 1.46, 1.82) and poor/very poor (OR = 2.17, 95% CI: 1.81, 2.60) emotional health (Table 3).

Older women were less likely to report worse emotional health, although this association was only statistically significant for fair emotional health (OR = 0.83, 95% CI: 0.72, 0.94) and poor/very poor emotional health (OR = 0.72, 95% CI: 0.57, 0.91) relative to very good emotional health. As was seen for QOL, women with higher levels of education were less likely to report worse emotional health (OR = 0.79, 95% CI: 0.69, 0.91 for good, OR = 0.70, 95% CI: 0.58, 0.84 for fair, and OR = 0.62, 95% CI: 0.43, 0.88 for poor/very poor emotional health, relative to very good emotional health). Women who were unemployed, on paid sick leave, students, or homemakers reported significantly worse emotional health relative to those who were employed (OR = 1.54, 95% CI: 1.11, 2.15 for good; OR = 2.75, 95% CI: 1.89, 3.99 for fair; and OR = 8.01, 95% CI: 4.53, 14.17 for poor/very poor emotional health, relative to very good emotional health). Women who did not report having a physician in charge of overseeing their follow-up care reported significantly worse emotional health (OR = 2.29, 95% CI: 1.29, 4.07 for good; OR = 2.38, 95% CI: 1.22, 4.67 for fair; and OR = 6.66, 95% CI: 2.89, 15.35 for poor/very poor emotional health, relative to very good emotional health) (Table 3).

## Discussion

Among breast cancer survivors in the Transitions Study, we found that with each additional reported chronic condition, participants were significantly more likely to report worse QOL and poorer emotional health. We also found that both overall QOL and emotional health were highest in women with higher educational attainment (more than a high school degree), those who had a physician in charge of their follow-up care, as well as those who were not unemployed or on paid sick leave.

The findings of our study may be explained by the additional physical and mental burden provoked by a breast cancer diagnosis among women already living with other chronic conditions. Prior research has also found that comorbid conditions can have a negative impact on health related QOL [28–31]. A recent prospective cohort study from the United States found that five years after breast cancer diagnosis, reporting one or more comorbid condition was associated with worse health related QOL scores [29]. Importantly, the study also concluded that poor health related QOL scores, were associated with a significantly increased hazard of all cause mortality [29]. Similarly, among breast cancer patients receiving chemotherapy, having a greater number of comorbid conditions has also been associated with poorer physical and role functioning, greater pain, worse sleep quality, and fatigue [30]. In a French cross-sectional study, having ≥2 comorbid conditions was associated with poorer QOL for the physical functioning and general health dimensions of the Short Form Health Survey (SF-12), but not the mental health dimension [31].

The impact of socioeconomic characteristics on QOL among breast cancer survivors is well documented [9, 32]. Prior research, and our current work have found that lower education levels and working as a homemaker or housewife are associated with worse QOL [33]. We also found that older women were less likely to report worse emotional health, consistent with prior work reporting poorer physical and psychosocial outcomes after breast cancer diagnosis among younger women [34, 35]. Further, women with breast cancer rely heavily on their physicians to provide information and support [36]. Positive communication with greater perceived self-efficacy in physician interactions is associated with better QOL [37]. Correspondingly, in our study, we found that not having a physician in charge of follow-up care was associated with worse QOL.

Chronic conditions can often worsen in severity and complexity over time, reducing individuals' functional status and thereby reducing QOL and emotional health. The manifestation of various chronic conditions, and their impact on functioning can vary—for instance—among cancer survivors, conditions such as osteoarthritis are linked to reduced physical functioning, whereas chronic psychological conditions including depression are strongly linked to emotional function [38]. The prevalence of multimorbidity—the co-occurrence of two or more chronic conditions—increases with age, and is greater among those with lower household incomes, and lower educational attainment [39]. According to a Danish study of cancer patients more broadly, comorbidity explained more of the variance in physical and emotional function components of health related QOL than sociodemographic characteristics and cancer characteristics (e.g. years since diagnosis, tumour stage, and treatment) [38]. Given the increasing prevalence of chronic conditions, and acknowledging that poor physical and mental health related QOL is associated with increased all-cause mortality [29], planning breast cancer survivorship care with chronic conditions in mind remains highly important.

Having breast cancer also negatively impacts adherence to chronic disease medications, and fewer primary care provider visits among survivors are associated with higher odds of non-adherence [40]. This may exacerbate already poor QOL among breast cancer survivors with a high comorbidity burden. In Canada, where healthcare services are delivered at the provincial level, adherence to guidelines for quality follow-up care of breast cancer survivors varies widely between provinces [41]. This is especially true for the management of chronic conditions among breast cancer survivors, where British Columbia has much lower levels of compliance relative to Ontario and Nova Scotia [41]. Ensuring quality follow-up care that is compliant with Canadian guidelines can serve as a target for intervention in Canadian provinces where compliance is low e.g. British Columbia [41].

## Strengths and limitations

This study has numerous strengths. First, the identification of study participants through provincial cancer agencies/registries, allowed for a source population of all individuals diagnosed with cancer within the province. Further, the Transitions Study survey development process included consultations with cancer survivors, clinicians and system leaders, as well as cognitive interviews with 15 cancer survivors to evaluate the questionnaire's meaningfulness, clarity, understandability, and ease of completion, thereby limiting potential information bias [19]. The survey also underwent performance testing with 96 survivors, who were recruited to match the eligibility criteria, further reducing the potential for information bias [19]. In addition, the overall QOL measure included in the Transitions Study questionnaire is comparable the global QOL measure from the EORTC QLQ-C30, which has been validated for use in breast cancer patients [25–27]. We also attempted to capture the impact chronic conditions on emotional functioning, one of the core domains of the QLQ-C30 by including emotional health as a separate outcome in our analysis. Despite the use of self-report survey data, there is limited potential for differential recall bias as all participants in the sample are breast cancer survivors. In addition, while the data are cross-sectional, temporality can be established because the exposure (number of chronic conditions) is assessed at the time of breast cancer diagnosis—albeit retrospectively, and the outcomes (QOL and emotional health) are assessed after completion of treatment. Finally, participants were selected through probability sampling, with all eligible survivors surveyed in smaller provinces [19]. However since survey weights were not available, the findings may not be generalizable to all Canadian breast cancer survivors [16, 19].

Limitations of this study include the lack of data on several potentially important predictors of QOL (e.g., menopausal status, and cancer stage at diagnosis). While a 30% response rate was assumed by CPAC when designing the survey and the response rate was 33%, this does not preclude the potential for selection bias in the study sample. In addition, the lack of information on individuals who did not respond to the questionnaire hinders the assessment of the extent of selection bias.

## Conclusion

Here we report that a higher burden of comorbidity was associated with worse QOL and poorer emotional health in a nationally representative sample of Canadians breast cancer survivors from the Transitions Study. These findings emphasize the importance of integrating information on chronic comorbid conditions into survivorship care to improve QOL and emotional outcomes of breast cancer survivors.

## Supporting information

**S1 File. STROBE checklist for cross-sectional studies.**
(DOCX)

**S1 Table. Coding details of all variables included in Table 1, and corresponding questions, response options, and variable codes from the Canadian Partnership Against Cancer (CPAC) Transitions Study survey.**
(DOCX)

**S2 Table. Unadjusted estimates of the relationship between the number of chronic conditions at breast cancer diagnosis, and the study outcomes: (i) quality of life, and (ii) emotional health, among breast cancer survivors in the Transitions Study.**
(DOCX)

## Acknowledgments

This analysis is based on the Canadian Partnership Against Cancer's Experiences of Cancer Patients in Transition Study survey. The Transitions Study was made possible by funding provided to the Canadian Partnership Against Cancer by Health Canada. The assumptions and/or calculations underlying the results were prepared by Jasleen Arneja, and the responsibility for the use and interpretation of these data and their reporting is entirely that of the author.

## Author Contributions

**Conceptualization:** Jasleen Arneja, Jennifer D. Brooks.

**Data curation:** Jasleen Arneja.

**Formal analysis:** Jasleen Arneja.

**Funding acquisition:** Jennifer D. Brooks.

**Methodology:** Jasleen Arneja, Jennifer D. Brooks.

**Resources:** Jennifer D. Brooks.

**Supervision:** Jennifer D. Brooks.

**Writing – original draft:** Jasleen Arneja.

**Writing – review & editing:** Jasleen Arneja, Jennifer D. Brooks.

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
