## [Decision Letter · Decision Letter 0]

23 Jun 2021

PONE-D-21-03223

The impact of chronic comorbidities at the time of breast cancer diagnosis on quality of life, and emotional health following treatment

PLOS ONE

Dear Dr. Brooks,

Thank you for submitting your manuscript to PLOS ONE. After careful consideration, we feel that it has merit but does not fully meet PLOS ONE’s publication criteria as it currently stands. Therefore, we invite you to submit a revised version of the manuscript that addresses the points raised during the review process.

We look forward to receiving your revised manuscript.

Kind regards,

David Teye Doku

Academic Editor

PLOS ONE

Journal Requirements:

3. In your Methods section, please provide additional information about the participant recruitment method and the demographic details of your participants. Please ensure you have provided sufficient details to replicate the analyses such as a) the recruitment date range (month and year), b) a description of how participants were recruited.

4. In the ethics statement in the manuscript Methods and the online submission form, please state whether you had access to any identifying information associated with the data, or if they were fully anonymised before access.

In addition, please state whether you were involved in any aspect of the original study including study design, data collection, analysis or manuscript preparation, or whether the data were exclusively obtained from a publicly available source

Reviewers' comments:

Reviewer's Responses to Questions

**Comments to the Author**

1. Is the manuscript technically sound, and do the data support the conclusions?

Reviewer #1: Yes

Reviewer #2: Yes

Reviewer #3: Yes

2. Has the statistical analysis been performed appropriately and rigorously? 

Reviewer #1: Yes

Reviewer #2: Yes

Reviewer #3: Yes

3. Have the authors made all data underlying the findings in their manuscript fully available?

Reviewer #1: Yes

Reviewer #2: Yes

Reviewer #3: Yes

4. Is the manuscript presented in an intelligible fashion and written in standard English?

Reviewer #1: Yes

Reviewer #2: Yes

Reviewer #3: Yes

5. Review Comments to the Author

Reviewer #1: Dear Editor,

Thank you for the opportunity to review the manuscript entitled “The impact of chronic comorbidities at the time of breast cancer diagnosis on quality of life, and emotional health following treatment”. This is a well written paper with great importance. I only have a few suggestions.

1. It will be great to add the study setting to the topic so readers can easily know where the study was conducted

2. I will also urge the authors to follow the STROBE Checklist and attach it as a supplementary file.

3. Line 21…"Using cross-sectional survey…" I think the authors can move this to the methods section of the abstract

4. Line 35 reads introduction while in the abstract(line 12) you have background, please make them consistent

5. Line 97…”prior literature…”please provide references to support this

6. It will be prudent for the authors to create sub-sections for the strength and limitations as well as the conclusion

Reviewer #2: I enjoyed reading this manuscript which uses data on from 3,372 breast cancer survivors who participated in the Canadian Partnership Against Cancer (CPAC) Experiences of Cancer Patients in Transitions Survey to determine the impact of comorbidities on self-reported quality of life (QOL) and emotional health following a breast cancer diagnosis and treatment. The manuscript is well-written and when published will contribute significantly to the literature on chronic conditions. I have a few comments which I believe will make the manuscript even stronger if considered and incorporated by the authors

Abstract

• Line 21-22. I suggest the authors send “Using cross-sectional survey data from 3,372 breast cancer survivors who participated in the Transitions study,” back to the methods section of the abstract.

Background

• This section is okay and described in details.

Methods

• I suggest the authors adopt the STROBE checklist (https://www.strobe-statement.org/index.php?id=available-checklists) in reporting the study as this will ensure all possible missing pieces are fitted into the right places. This will make the paper even stronger in terms of value. The authors would have to then state at the beginning parts of the methods that they adopted STROBE in reporting their findings

• While the authors have tried to present the variables in the methods section, I suggest they provide a table which contains all the variables (both outcome and explanatory), how they were coded in the survey and the questions which were asked regarding each of them. Such a table will effectively take care of lines 142 – 150 for instance.

Results

• The percentage for Education in Table 1 is 100.1% instead of 100%. Authors should kindly check and effect any other such changes.

• A first look at the topic “… chronic comorbidities at the time of breast cancer diagnosis…” suggests that participants included in this study were individuals who had at least one chronic condition aside breast cancer at the time of diagnosis; thus, taking into consideration the question on line 149–150. I am, therefore, not clear regarding the inclusion of the attribute “O” under the variable “Number of chronic conditions” in Table 1.

Discussion

• The discussion could be strengthened even further. For instance, while I commend the authors for juxtaposing their findings to previous research, it is important to also discuss the possible reasons for the key findings made, especially from line 229–260.

• A few typos and grammatical issues identified are worth correcting through a further proofreading of the manuscript before resubmission.

• Considering the cross-sectional nature of the data used for the analysis, I suggest the include the associated limitations.

Reviewer #3: Dear Author

thanks for the good paper looking at the association of number of co-morbidities with QOL

Do you have data on patients who refused interview that you can compare with those who accepted to rule out bias (there was less than 30% Response)

Any reason you choose to present co-morbidities other than. variables with high OR such as 'unemployment, having a doctor and age?

was this study approved by ethics committee? If yes please provide list of members. if NO, please state if t was 'exempt,.

The table on income was it for those currently working or did it include pension/ retirement benefit?

6. PLOS authors have the option to publish the peer review history of their article (what does this mean?). If published, this will include your full peer review and any attached files.

Reviewer #1: **Yes: **Abdul-Aziz Seidu

Reviewer #2: No

Reviewer #3: **Yes: **Dr. Evans Amukoye

---

## [Author Response · Author response to Decision Letter 0]

29 Jul 2021

Dear Dr. David Teye Doku,

Thank you for the opportunity to revise our manuscript, PONE‐D‐21‐03223, entitled, “The impact of chronic comorbidities at the time of breast cancer diagnosis on quality of life, and emotional health following treatment.” The Reviewers provided a thoughtful review and we believe the manuscript is improved after addressing their comments. Please see below for a point-by-point response to each comment/question. All edits to the manuscript are indicated using tracked changes and a clean version of the manuscript is also included. All line numbers referenced in the response to reviewers below refer to the clean version.

I would like to thank you for considering our manuscript.

Sincerely,

Jennifer Brooks

Assistant Professor

Dalla Lana School of Public Health, University of Toronto

Response to Journal Requirements

JR1. Please ensure that your manuscript meets PLOS ONE's style requirements, including those for file naming. The PLOS ONE style templates can be found at

Response: Thank you, we have ensured that our manuscript meets the style requirements.

JR2. Please review your reference list to ensure that it is complete and correct. If you have cited papers that have been retracted, please include the rationale for doing so in the manuscript text, or remove these references and replace them with relevant current references. Any changes to the reference list should be mentioned in the rebuttal letter that accompanies your revised manuscript. If you need to cite a retracted article, indicate the article’s retracted status in the References list and also include a citation and full reference for the retraction notice.

Response: Thank you, we have double checked our reference list. Our manuscript does not include any retracted papers. Changes to our references list include the addition of the following four references:

4. DuMontier C, Clough-Gorr KM, Silliman RA, Stuck AE, Moser A. Health-Related Quality of Life in a Predictive Model for Mortality in Older Breast Cancer Survivors. Journal of the American Geriatrics Society. 2018;66(6):1115-22.

21. Strengthening the Reporting of Observational Studies in Epidemiology (STROBE) Statement: Guidelines for Reporting Observational Studies. PLOS Medicine. 2007;4(10):e296.

22. Montazeri A, Vahdaninia M, Harirchi I, Ebrahimi M, Khaleghi F, Jarvandi S. Quality of life in patients with breast cancer before and after diagnosis: an eighteen months follow-up study. BMC cancer. 2008;8(1):330.

23. Mokhatri-Hesari P, Montazeri A. Health-related quality of life in breast cancer patients: review of reviews from 2008 to 2018. Health and quality of life outcomes. 2020;18(1):338.

JR3. In your Methods section, please provide additional information about the participant recruitment method and the demographic details of your participants. Please ensure you have provided sufficient details to replicate the analyses such as a) the recruitment date range (month and year), b) a description of how participants were recruited.

Response: The data used in our study come from the Experiences of Cancer Patients in Transitions Study (Transitions Study) and were previously collected by the Canadian Partnership Against Cancer (CPAC). We were not involved in the recruitment of participants in any way, and only had access to data that is available in the public domain. We have also referenced two papers published by CPAC-affiliated scientists, (references 19 and 20) that provide more in-depth information about recruitment. We were not able to provide the month of recruitment as that information was not available to us, however, we have provided the year of recruitment, and the range of weeks (8-19) it took to recruit participants in the 10 Canadian provinces. Additional information on participant recruitment is summarized in our manuscript (lines 78-91). Demographic characteristics of the study population are found in Table 1. 

JR4. In the ethics statement in the manuscript Methods and the online submission form, please state whether you had access to any identifying information associated with the data, or if they were fully anonymised before access.

In addition, please state whether you were involved in any aspect of the original study including study design, data collection, analysis or manuscript preparation, or whether the data were exclusively obtained from a publicly available source

Response: Anonymized data were used in this study. We have added to the Methods section that the data used in this study are in the public domain and freely available through the CPAC website. The authors were not involved in any aspect of the original study and did not have access to any identifying information associated with the data (lines 90-92). The authors exclusively obtained the data from the CPAC website.

Response to Reviewer Comments

Reviewer 1

R1.1 It will be great to add the study setting to the topic so readers can easily know where the study was conducted

Response 1.1 Thank you for this suggestion, we have added “in Canada” to the title.

R1.2 I will also urge the authors to follow the STROBE Checklist and attach it as a supplementary file.

Response 1.2 Thank you for this suggestion, we have completed a STROBE checklist and attached it as a supplementary file, and added this to the Methods section (line 93-94) As required in the STROBE guidelines, we have also provided unadjusted estimates of our results, available in S2 Table for both outcomes (quality of life and emotional health).

R1.3 Line 21…"Using cross-sectional survey…" I think the authors can move this to the methods section of the abstract

Response 1.3 Thank you, we have removed the sentence from the Results section of the abstract and identified the survey as cross-sectional in the Methods section of the abstract.

R1.4 Line 35 reads introduction while in the abstract (line 12) you have background, please make them consistent

Response 1.4 Thank you for pointing that out, we have changed “Background” to “Introduction” in the Abstract.

R1.5 Line 97… “prior literature…”please provide references to support this

Response 1.5 Thank you, we have added references to support our point that the prior literature largely uses health related quality of life (line 112).

R1.6 It will be prudent for the authors to create sub-sections for the strength and limitations as well as the conclusion

Response 1.6 We have now created a Strengths and Limitations sub-section and a Conclusion sub-section to the Discussion.

Reviewer #2:

Abstract

R2.1 Line 21-22. I suggest the authors send “Using cross-sectional survey data from 3,372 breast cancer survivors who participated in the Transitions study,” back to the methods section of the abstract.

Response 2.1 Thank you, we have removed the sentence from the Results section of the abstract and identified the survey as cross-sectional in the Methods section of the abstract.

Background

R2.2 This section is okay and described in details.

Response 2.2 Thank you.

Methods

R2.3 I suggest the authors adopt the STROBE checklist (https://www.strobe-statement.org/index.php?id=available-checklists) in reporting the study as this will ensure all possible missing pieces are fitted into the right places. This will make the paper even stronger in terms of value. The authors would have to then state at the beginning parts of the methods that they adopted STROBE in reporting their findings

Response 2.3 Thank you for this suggestion. We have now completed the STROBE checklist and included it as a supplementary file. This information is included in the Methods section (line 93-94). As required in the STROBE guidelines, we have also provided unadjusted estimates of our results, available in the S2 Table for both outcomes (quality of life and emotional health).

R2.4 While the authors have tried to present the variables in the methods section, I suggest they provide a table which contains all the variables (both outcome and explanatory), how they were coded in the survey and the questions which were asked regarding each of them. Such a table will effectively take care of lines 142 – 150 for instance.

Response 2.4 Thank you for this suggestion. We have included a table describing all the variables (exposure, outcome, and covariates) as a supplementary table (S1 Table). We have noted this in the Methods section (Statistical Analysis subsection line 136) as well as in the footnotes to Table 2 and 3 (lines 167 and 189, respectively).

Results

R2.5 The percentage for Education in Table 1 is 100.1% instead of 100%. Authors should kindly check and effect any other such changes.

Response 2.5 The percentages may not always add up to 100% due to rounding. We have added a footnote to clarify this, stating “Percentages may not add up to 100% in some cases due to rounding.”(line 153).

R2.6 A first look at the topic “… chronic comorbidities at the time of breast cancer diagnosis…” suggests that participants included in this study were individuals who had at least one chronic condition aside breast cancer at the time of diagnosis; thus, taking into consideration the question on line 149–150. I am, therefore, not clear regarding the inclusion of the attribute “O” under the variable “Number of chronic conditions” in Table 1.

Response 2.6 We thank the reviewer for noting this. We report in the abstract and in the Results section that 57% of the participants report at least one chronic condition, which implies that 43% of the population has 0 chronic conditions. On lines 103-106 also provide details as to the definition of the “number of chronic conditions” variable. We did not have any inclusion criteria based on number of comorbidities, but rather are investigating the impact of having comorbidities on QOL and emotional health. Individuals without comorbidities are used as a comparison group. 

Discussion

R2.7 The discussion could be strengthened even further. For instance, while I commend the authors for juxtaposing their findings to previous research, it is important to also discuss the possible reasons for the key findings made, especially from line 229–260.

Response 2.7 Thank you this suggestion. We have discussed possible explanations for our findings on lines 220-221.

R2.8 A few typos and grammatical issues identified are worth correcting through a further proofreading of the manuscript before resubmission.

Response 2.8 Thank you, we have proofread the manuscript and hope to have corrected all typographical and grammatical errors.

R2.9 Considering the cross-sectional nature of the data used for the analysis, I suggest the include the associated limitations.

Response 2.9 Thank you for noting this. We have addressed the use of cross-sectional data in our Strengths and Limitations section (lines 279-282). We do not believe that the use of cross-sectional data in this case is a limitation because temporality can be established because the exposure (number of chronic conditions) is assessed at the time of breast cancer diagnosis—albeit retrospectively, and the outcomes (QOL and emotional health) are assessed after completion of treatment.

Reviewer 3: 

R3.1 Do you have data on patients who refused interview that you can compare with those who accepted to rule out bias (there was less than 30% Response)

Response 3.1 Thank you for this question. We do not have information on individuals who did not complete the questionnaire. The survey was conducted by the Canadian Partnership Against Cancer (CPAC), and we have provided a reference to their published paper (Shakeel et al 2020), where they state: “A total of 40 790 survey packages were mailed across the 10 provinces and 13 319 responses were received (response rate = 33%); 12 929 surveys were completed by survivors aged 30 years or older.” A 30% response rate was assumed by CPAC when designing the survey. This does not preclude the potential for selection bias in the study sample and this has now been noted as a potential limitation (line 287-291)

R3.2 Any reason you choose to present co-morbidities other than. variables with high OR such as 'unemployment, having a doctor and age?

Response 3.2 Thank you for this question. The purpose of our study was to determine the relationship between the burden of comorbidity and quality of life (QOL) and emotional health. Since our primary exposure was the burden of comorbidity (as captured by the variable “number of chronic conditions”) we presented the corresponding results. The other variables included in the model (age, education, household size, employment status, marital status, and having a physician in charge of follow up) have previously been established as risk factors for poor QOL and poor emotional health in the literature and have been included in our analysis as potential confounders.

R3.3 was this study approved by ethics committee? If yes please provide list of members. if NO, please state if t was 'exempt,.

Response 3.3 This study was exempt from Research Ethics Board review as it was conducted with publicly available anonymized data with no personally identifiable information. We have included a statement to this effect (lines 94-95).

R3.4 The table on income was it for those currently working or did it include pension/ retirement benefit?

Response 3.4 The income distribution in Table 1 includes the entire sample regardless of employment status (employed, unemployed, homemaker, student, on paid sick leave, retired). This is now indicated in footnote ‘c’ for S1 Table which provides details on the coding of all variables. Please also note that we did not include the income variable in our models (also indicated in the same footnote).

---

## [Editor Report · Decision Letter 1]

10 Aug 2021

The impact of chronic comorbidities at the time of breast cancer diagnosis on quality of life, and emotional health following treatment in Canada

PONE-D-21-03223R1

Dear Dr. Brooks,

We’re pleased to inform you that your manuscript has been judged scientifically suitable for publication and will be formally accepted for publication once it meets all outstanding technical requirements.

Kind regards,

David Teye Doku

Academic Editor

PLOS ONE
---

## [Editor Report · Acceptance letter]

19 Aug 2021

PONE-D-21-03223R1 

The impact of chronic comorbidities at the time of breast cancer diagnosis on quality of life, and emotional health following treatment in Canada 

Dear Dr. Brooks:

I'm pleased to inform you that your manuscript has been deemed suitable for publication in PLOS ONE. Congratulations! Your manuscript is now with our production department. 

Kind regards, 

on behalf of

Dr. David Teye Doku 

Academic Editor

PLOS ONE